# The Landscape of Chronic Pain: Broader Perspectives

**DOI:** 10.3390/medicina55050182

**Published:** 2019-05-21

**Authors:** Mark I. Johnson

**Affiliations:** Centre for Pain Research, School of Clinical and Applied Sciences, City Campus, Leeds Beckett University, Leeds LS1 3HE, UK; M.Johnson@leedsbeckett.ac.uk

**Keywords:** chronic pain, analgesia, pain perception, sensory illusions, embodied pain, painogenic environment, evolutionary mismatch hypothesis

## Abstract

Chronic pain is a global health concern. This special issue on matters related to chronic pain aims to draw on research and scholarly discourse from an eclectic mix of areas and perspectives. The purpose of this non-systematic topical review is to précis an assortment of contemporary topics related to chronic pain and its management to nurture debate about research, practice and health care policy. The review discusses the phenomenon of pain, the struggle that patients have trying to legitimize their pain to others, the utility of the acute–chronic dichotomy, and the burden of chronic pain on society. The review describes the introduction of chronic primary pain in the World Health Organization’s International Classification of Disease, 11th Revision and discusses the importance of biopsychosocial approaches to manage pain, the consequences of overprescribing and shifts in service delivery in primary care settings. The second half of the review explores pain perception as a multisensory perceptual inference discussing how contexts, predictions and expectations contribute to the malleability of somatosensations including pain, and how this knowledge can inform the development of therapies and strategies to alleviate pain. Finally, the review explores chronic pain through an evolutionary lens by comparing modern urban lifestyles with genetic heritage that encodes physiology adapted to live in the Paleolithic era. I speculate that modern urban lifestyles may be painogenic in nature, worsening chronic pain in individuals and burdening society at the population level.

## 1. Introduction

Chronic pain is a global health concern with evidence that patients are receiving inadequate care, due in part to deficits in knowledge and skills of health care professionals [1,2]. Chronic pain is a biopsychosocial phenomenon, yet pain education for health professionals continues to focus on biomedical aspects of pain [3]. The purpose of this topical review is to précis an assortment of contemporary issues related to chronic pain and its management to reveal the landscape of current knowledge and thinking in the field. My intention is to bring certain issues to light through commentary rather than comprehensive in-depth objective appraisal of research literature. The review is narrative in style and I have based the direction and content of the review on what I find interesting and controversial. I have attempted to integrate knowledge from a variety of disciplines including philosophy, phenomenology, epidemiology, biomedicine, psychology, evolutionary biology and health promotion. My approach is ambitious. Arguments are anthropomorphic in nature and cannot be generalized to non-human species. I appreciate that the non-systematic approach to review is vulnerable to selection and evaluation biases and opinion-oriented arguments, so I direct readers to key references for comprehensive coverage of topics of further interest. I hope that some of the issues discussed in the review prompts scholarly debate about future directions for research, practice and health care policy.

## 2. The Paradox of Pain


*“To live in pain is not only to suffer aversive sensations but to be caught in a web of paradoxes”*
Leder [4], p. 2.

### 2.1. Defining Pain

Historically, clinicians viewed pain as a secondary symptom of injury and disease and focused treatment on removing the precipitating (primary) cause. Relief of pain was of secondary concern. Nowadays the importance of alleviating pain is widely accepted. The International Association for the Study of Pain (IASP) defines pain as: ‘*An unpleasant sensory and emotional experience associated with actual or potential tissue damage, or described in terms of such damage.*’ [5]. The association between tissue damage and pain is at the core of the definition, although the clause ‘… *or described in terms of such damage*’ avoids *always* binding pain to the stimulus (i.e., potential or actual tissue damage). This is because pain is a psychological state produced by the brain in response to a multitude of biopsychosocial inputs of which activity in nociceptive (noxious detecting) pathways is but one. Often, scientists and clinicians view pain as a technical problem solvable by biomedical interventions (e.g., drugs and surgery) that target physiology, pharmacology, biochemistry, and molecular biology of the nociceptive system. This approach has proven successful at advancing our understanding of the pathophysiology of pain and identifying novel biomedical targets to alleviate pain. The focus on a biomedical approach has been at the expense of the contribution of psychosocial and environmental factors in the lived experience of pain. This may be one reason why management of chronic pain remains a challenge and the burden of chronic pain on society continues to rise. Pain is complex.

### 2.2. The Lived Experience of Pain

The lived experience of pain is a perplexing mix of sensory, emotional and cognitive phenomenon that fluctuate in and out of conscious awareness. The unpleasant nature of pain demands attention, explanation and action to resolve actual or potential damage of the body. An assortment of biopsychosocial and environmental factors influence the appearance, severity, character, and time course of pain, although often pain is unpredictable. Pain may occur in response to noxious stimuli, innocuous stimuli or in the absence of any apparent stimuli. From a phenomenological perspective people report their experience of pain to fluctuate in the present and the past and the future; to be localized and radiating everywhere; to be productive and destructive of value and meaning; and to be never changing and ever-changing [4]. Often pain is amorphous.

### 2.3. Why Do We Communicate Pain?

Expression of pain is at the core of human group dynamics, serving to inform other individuals that you are injured or ill. In modern society expressing pain has the potential to generate empathy in others to motivate them to offer aid. Individuals evaluate a person’s expression of pain to determine the extent of their disability and whether the person can fulfill their responsibilities. It seems plausible that expression of pain was beneficial for our human ancestors living as hunter-gatherers in the Paleolithic era [6]. Hunter-gatherer groups need to keep moving to search for food and avoid predation. Providing assistance or care to group members who were injured or ill could be detrimental to a group’s ability to acquire sufficient food or remain safe. Thus, the ability to legitimize pain to others would have been important when trying to persuade others to provide care and assistance. Interestingly, individuals could use this to their advantage by claiming to be in pain when in fact they were not, to avoid dangerous tasks such as hunting trips. Legitimizing pain to others is particularly challenging in the absence of visible cues of injury or illness because other group members may believe that the person is expressing pain to seek an unfair advantage. It seems plausible that evolution has hardwired the nervous system to exaggerate expression of pain (to be believed) and paradoxically hardwired the nervous system to be skeptical about the existence of pain in others (to prevent being duped). This paradox plays out in the adversary struggle that chronic pain patients experience when trying to legitimise their pain in the health care system [7,8,9,10].

## 3. The Struggle to Legitimize Pain


*“To have pain is to have certainty; to hear about pain is to have doubt”*
Scarry [11], p. 13.

### 3.1. Disproving Pain

Communicating the complex, dynamic, and multidimensional nature of pain experience is a challenge. The amorphous character of pain does not sit comfortably with the objective nature of medical practice and evidence suggests that chronic pain patients have difficulties convincing health care professionals of the existence of pain [7,9,10]. Pain is a perceptual experience that is personal to the individual and by definition unobservable by another person. The subjective nature of pain makes it is impossible to prove or disprove a person’s pain and therefore it is not possible to distinguish a person’s report of pain experience from that of tissue damage. A person’s report of pain should be accepted if their sensory and emotional experience of pain is expressed in the same ways as that caused by tissue damage. This was recognised in 1968 by McCaffery who defined pain as ‘…*whatever the experiencing person says it is, existing whenever the experiencing person says it does*’ [12]. Hence, conveying pain experience depends on an ability to use language and/or behavioral action to communicate the internal state of one’s body. It also depends on an ability to persuade others of the existence of pain in oneself. Likewise, the observer needs to be receptive to what is being communicated, including being able to interpret the meaning and importance of both verbal and behavioral information. This can be challenging for individuals with limited verbal expression such as infants or with compromised cognitive ability such as individuals with dementia. A recent systematic review of pain management for community-dwelling people with dementia found that informal caregivers were more likely to report pain on behalf of the person with dementia and pain-focused behavioral observation assessment was infrequently used by practitioners [13]. Multimodal assessment of pain that includes self-reported and non-self-reported measures have been developed to capture the complexity of pain experience [14], although it is important to note that behavioral signs, effort testing, self-reported questionnaires, or symptom validity tests have been shown to be unable to identify malingering [15].

### 3.2. The Tenuous Link between Pain and Pathology

Skepticism about the legitimacy of another person’s pain is at its height when there is an absence of evidence of injury or disease [16]. For example, pain may be driven by central sensitization and/or abnormal functioning of the nociceptive system, and/or altered higher level processing such as fear-avoidance behavior. Unlike many illnesses, pain is not visible or measurable using objective means and clinicians have to rely on patients’ self-report, coupled with observation. Subjective and objective assessment, including the patient’s report of symptoms, examinations of functional capacity, and diagnostic tests, contribute to pain diagnoses. When legitimizing pain and constructing logical explanations for pain, patients and non-pain specialist practitioners tend to give credence to positive diagnostic tests related to tissue pathology, at the expense of alterations in physiological processing associated with pain se (e.g., central sensitisation). Diagnostic imaging techniques are critical for detecting sinister pathology in patients presenting with pain. However, tissue pathology may be present in the absence of pain [17] so paradoxically, evidence of tissue pathology may be counterproductive when searching for causes of pain in some circumstances Thus, in some instances the relationship between pain, injury and disease may be tenuous.

Pain may occur in the absence of injury. A clinical anecdote by Fisher et al. [18] described the case of a builder who presented to accident and emergency complaining of severely disabling pain due to a 15-cm nail that had penetrated his boot. The builder needed strong analgesia and sedation before physicians were able to remove the nail from the boot. On removing the boot from the foot, they discovered that the nail had lodged between the toes without causing tissue damage! Pain may occur in the presence of minor injuries. A sliver of metal embedded in the skin of a finger is a good example. Paradoxically, serious injuries may be devoid of pain. Beecher et al.’s seminal paper reported that over 50% of soldiers with fresh combat wounds reported mild or no pain [19]. Evidence suggests that high proportions of individuals without pain have pathological changes associated with aging [17]. Clinical care pathways for some chronic pain conditions do not recommend diagnostic imaging in the absence of red flags (e.g., non-specific low back pain [20]). Pathology may not always be driving pain.

### 3.3. Pain is Not a Unitary Phenomenon

Clearly, the dynamic, multidimensional, and amorphous nature of pain is challenging to capture with any degree of specificity and precision. Commonly, pain is reduced to a unitary phenomenon measured by numerical intensity rating scales. This approach is convenient and psychometric research evidence suggests that data gathered from scales is valid and reliable [21,22,23]. However, rating scales do not measure pain objectively and can give a false impression of the level of measurement precision (e.g., 1 mm on 100-mm visual analogue scale). Scales presume linearity of subjective report between scale ends and use anchors that are nebulous (e.g., ‘*The worst-ever pain’* or ‘*The worst pain imaginable’*). Patients have been known to extend scale ends to incorporate incidents that have provoked pain beyond the worst they had previously thought imaginable [24]. The measurement of pain in this way is not only imprecise but is also fails to capture the complex and subjective nature of pain. To overcome the restrictions associated with numeric scaling Wideman et al. [14] have offered a multimodal assessment model of pain that describes quantitative aspects of pain such as self-reported and non-self-reported measures, and importantly, qualitative aspects of pain such as the words and behaviors of the patient’s narrative. This practical framework assists the integration of the subjectivity of pain within assessment. The inclusion of, and importance paid to, the narrative report captures more fully an individual’s pain experience helping to legitimize pain for both patient and practitioner. This promotes a compassion-based consultation and provides much information that enriches assessment of the underlying processes contributing to pain.

## 4. Chronic Pain

### 4.1. Reassessing the Acute–Chronic Dichotomy?

Traditionally, service delivery and clinical practice views pain through an acute or chronic lens. Acute pain is pain persisting less than twelve weeks. Chronic pain is often secondary to disease or traumatic injury and initially considered a symptom. Chronic pain is pain that persists or recurs for twelve weeks or more, or beyond the expected time for healing. Recently, Loser argued that the acute–chronic dichotomy is so entrenched in pain parlance that it has escaped critical scrutiny [25]. There are no temporal correlates of physiological processes associated with pain based on time points used to distinguish acute and chronic. Loser argues that we should describe pain syndromes based on physiological mechanisms, including peripheral or centrally generated perspectives as originally discussed by John Bonica in the 1950s.

From an evolutionary perspective, hypersensitivity of the nervous system serves to assist the healing process by amplifying and prolonging pain. This discourages use of, and contact with, injured tissue. Peripheral sensitization is driven by the release and production of biochemical mediators at the site of tissue damage that lower the threshold of activation of transducer ion channel receptors expressed in nociceptor terminals. Sometimes the adaptive function of peripheral sensitization is lost as is the case for some autoimmune diseases such as rheumatoid arthritis that generates ongoing inflammation, peripheral sensitization and pain hypersensitivity even though healing does not occur.

Central sensitization is centrally mediated amplification of pain irrespective of mechanism or location. Central sensitization is triggered by noxious (excitatory) input arising from direct activation of nociceptors (nociceptive) or from damaged or dysfunctional neuronal fibers (neuropathic). Nerve injury can reduce segmental and/or extrasegmental inhibitory influences on central nociceptive transmission (i.e., disinhibition) through a loss of interneurons, loss of descending pathways, altered connectivity and microglial activation. These mechanisms lower the threshold of excitation of central nociceptive transmission neurons, amplifying their output to noxious and non-noxious stimuli [26]. The molecular mechanisms are multifactorial and complex with NMDA (N-Methyl D-Aspartate) receptors having a critical role, in a process similar to long-term potentiation associated with memory formation. The receptive fields of central nociceptive transmission neurons also expand so that they become responsive to stimuli applied to areas of tissue that do not normally activate them. Thus, central sensitization increases the area of pain hypersensitivity across body parts.

Central sensitization manifests primary and secondary hyperalgesia and allodynia and sometimes pain presents spontaneously in the absence of nociceptive stimuli. Patients may present with widespread pain in multiple body regions and pain arising after mundane activities such as walking or cooking. Yunus coined the term central sensitivity syndrome to describe pain-related conditions without obvious tissue pathology, that have similar comorbid symptoms (e.g., poor sleep hygiene, fatigue, and slowness of cognition) and include fibromyalgia and irritable bowel syndrome [27].

### 4.2. Nociplastic Pain: A New Mechanistic Descriptor?

Presently, mechanistic categories of pain are; *nociceptive pain*, resulting from activation of nociceptors by a noxious stimulus that is damaging or threatens damage to healthy tissues (other than neural structures); and *neuropathic pain* resulting from a lesion or disease of the somatosensory nervous system. Recently, Kosek et al. [28] have proposed consideration of a third mechanistic descriptor, *nociplastic pain* (other candidate terms offered were algopathic or nocipathic pain), to describe pain arising from altered central nociceptive processing in the absence of tissue damage. Kosek et al. argue that inclusion of nociplastic pain, or some other equivalent, in the vocabulary of pain would raise awareness that pain may present without detectable tissue damage and promote screening for signs of nociceptive dysfunction to improve diagnosis and treatment. Nociceptive and neuropathic mechanisms that contribute to pain are proven and can be detected using various techniques including biochemical investigations, radiology, nerve conduction tests and imaging of the nervous system. It is important to recognize that the terms nociplastic, nociceptive and neuropathic are not diagnoses or exclusive categorical labels but descriptors of concurrent potential mechanistic drivers of pain. Nevertheless, the term nociplastic pain could help patients create explanatory models of their pain experience to legitimize their pain to others.

### 4.3. The Burden of Chronic Pain

Most literature discusses pain from an acute–chronic dichotomy. Chronic pain affects between 15–30% of the general adult population [29,30,31], with severe, debilitating chronic pain affecting 10–15% of adults [32]. Healthcare and socioeconomic costs of chronic pain is high and estimated to be 3–10% of gross national domestic product in Europe [33]. In the United States annual costs related to health care delivery and lower worker productivity due to chronic pain is estimated to be between $560 and $635 billion dollars and greater than heart disease ($309 billion), cancer ($243 billion), and diabetes ($188 billion) [34].

People do not die directly of pain, and unlike functional impairment, pain is not visible. Pain is often secondary to other medical conditions. Health care policies often focus on curing or slowing progression of the primary disease as a means to improve functional outcome, and may neglect pain management. For example, pain is underdiagnosed and inadequately managed in neurological conditions [35,36], despite high prevalence (e.g., Parkinson’s disease (40–85% (range) [37]), multiple sclerosis (55–70% (95% confidence intervals (CI)) [38]), motor neuron disease (19–85% (95% CI) [36]) and Alzheimer’s disease (38–75% (95% CI) [36]). There has been a long-standing debate whether chronic pain should be considered a disease entity in its own right under certain circumstances [39].

### 4.4. International Classification of Diseases: Chronic Primary Pain

The 11th edition of the World Health Organization’s International Classification of Diseases (ICD-11) categorises chronic pain as secondary to other conditions: chronic cancer-related pain, chronic neuropathic pain, chronic secondary visceral pain, chronic posttraumatic and postsurgical pain, chronic secondary headache and orofacial pain, and chronic secondary musculoskeletal pain [40]. The ICD-11 recognises that in some circumstances, chronic pain may not be explained by the presence of another condition and require special treatment and care in its own right. Thus, ICD-11 includes a category for chronic primary pain to reflect in part that pain should be regarded as a pathologic entity in its own right and characterized by a dysfunctional nervous system with persistent central sensitization (for review of the history of this debate see Raffaeli, and Arnaudo [39]).

Chronic primary pain defies classical pathological based classification and is described as pain as the primary complaint in one or more regions of the body and causing significant emotional distress or functional disability. Specific examples include chronic widespread pain, fibromyalgia, irritable bowel syndrome and non-specific chronic low back pain. Chronic primary pain draws attention to the much broader spectrum of possible biological, psychological and social causes and consequences than tissue damage or on going disease. The premise that pain is a biosychosocial phenomenon is widely accepted but social components of pain are often absent from pain definitions. Recently, Williams and Craig have called for the IASP to re-consider its definition of pain to highlight the contribution of social elements in pain experience, as follows: ‘…*a distressing experience associated with actual or potential tissue damage with sensory, emotional, cognitive, and social components’* [41] (p. 2420). This would draw further attention to the need to deliver holistic models of care.

## 5. Desirable Models of Care

It is recommended that patients with chronic pain are managed using a biopsychosocial model of care with pharmacological and non-pharmacological interventions tailored to the needs of the individual [42]. Care plans should promote physical and psychological wellbeing through lifestyle adjustments (e.g., in diet and physical activity), psychological interventions (e.g., cognitive behavioral therapy), and non-pharmacological adjuncts such as manual therapies, transcutaneous electrical nerve stimulation (TENS), thermal therapies, acupuncture, low level laser therapy, mirror therapy and virtual reality. The World Health Organization promotes the use of multidisciplinary teams working in partnership with patients to co-create explanations about pain and construct care plans that empower individuals to be active participants in their treatment because it creates self-efficacy [42,43,44]. Traditionally, first line treatment for pain involves the use of analgesic and adjuvant drugs that modulate nociceptive system processing, and the simplicity, convenience and partial success of this approach has meant that it dominates service delivery in many parts of the world.

### 5.1. The Analgesic Ladder

The World Health Organization’s analgesic ladder, initially developed for cancer pain [45], advocates a stepwise approach to prescribing starting with mild analgesics and increasing dosage or switching to powerful analgesics if pain is not adequately managed. Paracetamol or nonsteroidal anti-inflammatory drugs (Non-Steroidal Anti-Inflammatory Drugs (NSAIDs), e.g., diclofenac and ibuprofen) are prescribed for mild to moderate non-neuropathic pain, stepping up to weak opioids (e.g., codeine and dihydrocodeine) for moderate pain, and strong opioids (e.g., morphine and oxycodone) for moderate to severe pain. Prescribers also use local anesthetics for mild pain (e.g., mouth ulcers), and for moderate to severe pain (e.g., during and post-surgery). Adjuvants are drugs whose primary use was not originally for relief of pain and are used to manage neuropathic pain (e.g., amitriptyline, duloxetine, gabapentin or pregabalin) or muscle spasm (e.g., baclofen).

Analgesic drugs interact with nociceptive pathways to inhibit the onward transmission of noxious information from the site of injury to the brain in order to alleviate pain. Non-steroidal anti-inflammatory drugs inhibit cyclooxygenase reducing the production of prostaglandins that normally sensitize nociceptors; opioids prevent onward transmission of nociceptive transmission in the central nervous system; and local anesthetics block sodium channels reducing transmission of nerve impulses in nociceptive fibers. Analgesic and adjuvant medications are extremely valuable for short-term management of pain and used cautiously in the long-term. The largest pan European survey of chronic pain found that 64% of participants stated that medication did not adequately control pain [2]. It may be difficult for some health care professionals to accept that long-term drug medication may not be the best option and that the multidimension nature of pain means that response to drug intervention is highly malleable. This malleability is demonstrated in research using open-hidden paradigms. A review of experimental evidence by Benedetti [46] provided evidence that the efficacy of analgesic drugs is modulated by cognitive and affective state. Administering a placebo (fake) drug coupled with the suggestion that it is an analgesic drug increases analgesia and administering an analgesic drug with the suggestion that it is a placebo (fake) drug reduces analgesic outcome. The addition of open-label placebo treatment as an adjunct to treatment for chronic lower back pain has been shown to be safe and effective at reducing pain raising interesting ethical debates about the role of placebos in treatment [47].

### 5.2. Overprescribing

Concerns have been expressed about detrimental consequences to individuals and society of over diagnosis and over prescribing for chronic long-term illnesses, including chronic pain [48,49]. Abuse of prescription opioids is a major problem in some Western countries [50], although paradoxically, restricted access to opioids impedes pain management in parts of Asia, Africa, and Middle East [51]. The use of prescription opioid medication in the United States has risen from 4.1% in 1999–2000 to 6.8% in 2013–2014, with long-term users rising from 45.1% of all opioid users in 1999–2000 to 79.4% in 2013–2014 [52]. Social, economic and environmental variables are as influential in the opioid epidemic as health and biomedical factors. Recently, cross-sectional analyses of individual and county-level demographic and economic factors have found that elevated mortality from opioid overdose due to increased opioid prescribing was positively associated with direct marketing of opioid medication to physicians [53], and that chronic opioid use was positively associated with presidential voting patterns in USA counties [54]. Abuse of prescription opioids demonstrates the need for greater caution and selectivity in prescribing of long-term opioid therapy. Ironically, as one drug loses favor others come into view.

### 5.3. Novel Centrally Acting Drugs

There is a debate whether plant-based cannabis preparations could replace opioids [55]. There is low-level evidence that cannabinoids alleviate neuropathic pain and insufficient evidence to recommend their clinical use for other types of pain [56,57]. Presently, professional bodies recommend that cannabinoids be considered in exceptional circumstances for neuropathic pain, chronic non-malignant pain and cancer pain when patients are not responding to other treatment [58]. Centrally acting psychedelic drugs such as lysergic acid diethylamide and psilocybin have also attracted interest for the management of chronic pain [59]. Psychedelics act as agonists at 5-HT_2A_ receptors and may modulate pain through action in the rostral ventromedial medulla enhancing descending pain inhibitory pathway activity [60]. Psychedelics influence metacognitive interpretation of pain including the resting state of awareness mediated by the default mode network in the brain [61]. Pain interferes with activity in the default mode network causing intrusive cognition and the breakdown of normal self [62]. Only a few small studies without controls exist that suggest potential benefit of psychedelics for cluster headache and malignant and neuropathic pain [59].

## 6. Service Delivery in Primary Care

The value of analgesic medication is undisputed providing it is within a biopsychosocial model of care. Adams and Turk [63] suggest that a biopsychosocial perspective, rather than a singular biomedical, or psychological, or social perspective, is necessary to fully understand the lived experiences of people with chronic central sensitivity pain syndrome. A comprehensive approach to assessment is optimal to address physical, emotional, and social functioning and a palliative approach is optimal to manage pain and distressing symptoms because there are no curative treatments. However, resource pressures often hinder the delivery of a truly biopsychosocial model of care by multidisciplinary teams in primary care settings.

### 6.1. Shared Appointments and First Points of Contact

Employing multidisciplinary teams of physicians, nurses, physiotherapists, psychologists, dietitians and occupational therapists can be financially costly. Traditionally, the first point of contact is the primary care (general) physician (GP) who conducts a short one-on-one ten-minute consultation with the patient to assess, diagnose, agree appropriate treatment and discuss difficult concepts. This consultation often feels rushed with GPs having similar consultations with different patients over the course of a day. Shared appointments may be a more cost-effective model of service delivery than one-to-one consultations and have been successful for the management of diabetes, and obesity [64]. Shared appointments involve each patient consulting with the clinical team, with elements of group education and discussion between patients, and peer support by learning from the experiences of other patients. Patients impart information more readily in shared appointments and engagement with patients with similar conditions provides motivation for health behavior change. There is increasing use of shared medical appointments for persistent pain [64]. Moreover, physiotherapists and nurses may be better suited than GPs as a first point of contact to diagnose and advise patients with non-complex chronic pain [65].

### 6.2. Social Prescribing

Some countries have adopted programmes of social prescribing to provide patients in primary care with sources of support within their community. Social prescribing enables healthcare professionals to refer patients to non-medical personnel who work with the patient to co-design a nonclinical social prescription to improve health and well-being [66]. Social prescriptions include access to practical information and advice, community activity, and physical activities provided by voluntary and community sectors. A systematic review of 15 evaluations of social prescribing schemes in UK settings could not reliably judge effectiveness or value for money because evaluations had a paucity of data and methodological limitations, although conclusions of individual evaluations were generally positive [67]. Social prescribing schemes seem to be an ideal fit for the management of many chronic pain conditions because social prescribing addresses psychological, social and environmental factors affecting health and well-being associated with pain.

## 7. Pain Perception: Active Top-Down Processing?

Classically, a ‘bottom-up’ stimulus-organism-response model describes the physiological processes involved in producing pain sensation. The model inadvertently implies that pain is an inevitable consequence of activity in the nociceptive system driven by tissue damage. This stimulus-organism-response model has been refined to incorporate changes in the sensitivity of the nociceptive system and top-down processes that facilitate and inhibit nociceptive transmission [68,69]. Melzack suggested that the multidimensional experience of pain resulted from characteristic patterns of nerve impulses (i.e., a neurosignature) produced within multiple widely distributed neural networks in the brain that are genetically determined and modified by sensory experience (i.e., body-self neuromatrix theory of pain) [70]. Melzack suggested that multiple factors influenced the output patterns of the body-self neuromatrix, of which noxious input was only one. This would explain why pain sometimes arises and/or persists despite limited noxious input.

### 7.1. Maleability of Perceived Properties of the Body

Predictions and expectations are a core feature of the central nervous system processing enabling the brain to generate perceptual experiences based on snippets of multisensory input. This ‘perceptual inference’ is involved in the generation of the sense of ownership of body parts, and in the location of sensory events, including pain, within the body. The perceptual qualities associated with the sense of body ownership are malleable as demonstrated by somatosensory illusions.

The rubber hand illusion demonstrates how rapidly the brain can update assumptions about the location of stimuli and ‘ownership’ of body parts. An individual watches a rubber hand stroked with a brush whilst their real hand is stroked in synchrony but hidden out of view [71]. Within minutes, the sensation of stroking feels as if it is arising from the rubber hand and the individual experiences a sense that the rubber hand has become part of their body (i.e., the rubber hand has been embodied and the real hand disembodied) [72]. Approaching a perceptually embodied rubber hand with a threatening stimuli may elicit somatosensations in some individuals including pain, subjective anxiety and skin conductance responses that are similar in magnitude to that experienced when a real hand is threatened [73,74].

The marble hand illusion demonstrates how the brain can update assumptions of the material qualities of the body. The illusion involves gently hitting the hand with a hammer and progressively replacing the sound of hammer blow on skin with a hammer blow on marble. Within five minutes the hand feels stiffer, heavier, harder, less sensitive and unnatural [75]. Protecting a virtual limb with a virtual iron armor cover can reduce electrically evoked pain [76] and hearing a creaky door when moving a stiff joint or back increases the need to protect the spine whereas hearing a gentle whoosh reduces the need to protect [77]. In fact, perceptual stiffness in the back was found not to be a representation of biomechanical properties of tissue but rather a perceptual inference error of stiffness, in other words, “… *feeling stiff does not equate to being stiff in chronic low back pain*” [77] [p2]. Thus, the brain operates to reduce threat and preserve coherent behavior according to situational context and these perceptual inferences can operate in both directions, i.e., pain or analgesia [78,79].

### 7.2. Multisensory Perceptual Inference as a Protector

Pain is never motivationally neutral. Pain is a potent driver of action in much the same way as thirst drives drinking and hunger drives eating, because the cost of ignoring pain may result in tissue damage, disrupted homeostasis and threat to life. Outcomes of behaviors that do not meet predictions (i.e., are unexpected with a large prediction error) have a major influence on future behavior. Unexpected pain, such as a severe lancinating shooting pain during an innocuous movement, is likely to have a disproportionately large effect on the expected intensity of future encounters with the same stimuli. Unexpected pain is likely to amplify multisensory perceptual inference serving to protect the integrity of tissue by creating, for example, fear-avoidance of movement (i.e., the motivational-decision model of pain [80]). Individuals experiencing pain over-estimate the distance and the effort needed to walk to a target [81] and perceive targets to be further away [82] (i.e., economy of action hypothesis), although potential scaling of spatial perceptions during pain have not been consistently demonstrated [83]. Actions that fail to restore coherence of behavior may also contribute to perceptual dysfunctions accompanying pain. Examples include phantom limbs stuck in one position [84], and painful limbs feeling excessively large in complex regional pain syndrome [85,86], or excessively small in osteoarthritis [87]. Contemporary models of pain perception are attempting to integrate sensory, affective, cognitive, social, and bodily cues interpreted within social, environmental and evolutionary contexts, including previous experiences, from the perspective of embodied cognition.

### 7.3. The Theory of Embodied Pain

The embodied theory of pain designates pain as arising from situations that *infer* bodily threat to drive behaviors to reduce the impact of the threat on the integrity of tissues (i.e., defensive) whilst maintaining the integrity of rational behavior [88]. The embodied theory describes pain as a dynamic, motor experience rather than a passive, sensory experience and blurs the distinction between perception and action: “*Pain is always about action*.” ([88], p. 3). The embodied theory of pain places the body and its ability to investigate the environment through a dynamic exchange of information between nervous system and environment at the core of pain experience. Thus, the brain uses information about previous encounters with pain and previous behaviors on pain to ‘flavour’ pain experience. For example, individuals report thermal stimuli to be hotter and more painful when delivered in synchrony with a red light (often associated with heat) compared with a blue light [89]. Appreciating that the brain uses multisensory perceptual inference to predict likely consequences of undertaking defensive behaviors offers opportunities for therapeutic interventions [88,90,91,92,93].

### 7.4. Multisensory Perceptual Inference to Alleviate Pain

Therapeutic opportunities can arise from manipulating the environment and context to reinstate coherence of behavior and normalize perception. Mirrors, lenses, and virtual reality have been used to alleviate pain and improve function through visual distraction, ‘normalizing’ the appearance of dysmorphic painful body parts, reducing threat associated with moving painful body parts, and aligning ownership and agency of body parts through visual, proprioceptive, and tactile congruency. Therapeutic success is varied and research findings from systematic reviews of controlled clinical studies are inconsistent due to a paucity of robust primary studies [94,95,96,97].

Advanced technologies that couple visual, auditory, and haptic (tactile) stimuli, such as vibration in hand held game controllers and force feedback systems for medical and military training are being adapted to manipulate the multisensory environment impinging on the body influencing embodied and embedded perceptual experience. Immersive virtual reality technology using head-sets and non-immersive virtual reality technology using computer screens can be used in combination with motion tracking systems so that movements of an individual’s real limb can be used to control the movement of a virtual limb. Pain may be modulated when individuals are immersed in different virtual reality environments [98,99]. Virtual and augmented reality technologies providing enriched practice environments tailored to individual needs are being used in recovery and rehabilitation after brain damage or injury to facilitate motor learning and neural plasticity. Virtual reality offers several opportunities for pain management including distraction from painful body parts, providing contexts that reduce perceived threat, resizing of painful dysmorphic body parts, re-embodiment of alienated and/or disowned painful body parts, and modulation of agency to facilitate movement of painful body parts accompanied by fear-avoidance of movement.

Task-oriented virtual reality provides opportunities to reduce the sense of threat associated with moving painful body parts accompanied by fear-avoidance of movement [100,101,102,103]. For example, Ortiz-Catalan et al. [104] designed a virtual environment as a rehabilitation training aid for individuals with amputated upper limbs. Surface electrodes were used to record muscle activation over the stump of the residual limb whilst individuals tried to voluntarily control a virtual phantom limb that was displayed in real-time on a computer screen. Motor volition was decoded using myoelectric pattern recognition software whilst the patient matched random target postures or attempted to drive a virtual car using the virtual limb. Improvements in phantom limb pain occurred after 12 training sessions that was sustained for six months post-training.

## 8. Social and Environmental Contexts

At the core of multisensory perceptual inference of pain is the social context in which an individual lives, including previous experiences associated with pain. Thus, an individual’s pain experience is embedded in environmental settings.

### 8.1. Lifestyle and Chronic Pain

Increasingly, modern human lifestyles are embedded in urban environments. Modern urban lifestyles are associated with mortality and morbidity of noncommunicable ‘lifestyle’ diseases, such as cardiovascular diseases (heart attacks and stroke), chronic respiratory diseases (chronic obstructed pulmonary disease and asthma), diabetes and cancer [105]. Chronic pain is a secondary consequence of many of these non-communicable diseases causing suffering, disability and a significant impact on quality of life. The Global Burden of Disease Study 2013, found that chronic low back pain had the highest number of years lived with disability [106]. Some lifestyle behaviours are risk factors for non-communicable diseases including sedentary activity, unhealthy diet, anxiety and depression, smoking, lack of sunshine, disrupted sleep, unemployment, living in adverse socioeconomic circumstances, and previous history of abuse or violence [107,108,109]. The relationship between these lifestyle factors and chronic pain is complex and unclear with causal processes likely to be acting in both directions.

Built and food environments are known to promote obesity in populations and has been described as obesogencity of urban environments [110]. The concept of an obesogenic environment has helped to focus attention upstream on whole systems public health solutions including the design of urban environments [111]. There are similarities between obesity and chronic pain [112]. Both conditions disproportionately affect poorer people society; are associated with high economic and social costs; are influenced by biopsychosocial factors including physical activity and diet; and are managed using pharmacological, educational and behavioural interventions. This raises the possibility that modern urban environments may be painogenic in nature [113].

### 8.2. Painogenicity of Modern Urban Living

Painogenicity is the sum of influences that the surroundings, opportunities or conditions of life have on promoting persistent pain in individuals or populations [113]. Modern urban environments generally consist of air polluted with toxic emissions and particulates, high population densities, limited green open space and readily available processed foodstuffs. Consequently, there has been a shift toward sedentary lifestyles and high calorie diets of over processed food with excess sucrose, salt, fat and additives, both known to contribute to noncommunicable ‘lifestyle diseases’ (c.f. Sick City Syndrome, Modern Urban Living Syndrome) [114]. The shift to modern urban living is more rapid than physiological adaptation resulting in a potential evolutionary mismatch [114,115]. An evolutionary lens has been used to explore the biological role of sensitization and hyperalgesia in chronic pain [78,116,117,118] but little is written about the contribution of evolutionary mismatch to chronic pain [113].

### 8.3. Chronic Pain and Evolutionary Mismatch

Evolutionary mismatch may provide insights to potential painogencity of modern urban environments. Our genetic heritage encodes physiology adapted for hominin ancestors that existed in the Paleolithic era, circa 4–7 million years ago with clean air, exposure to microorganisms and lifestyles consisting of walking, climbing, lifting, carrying and bending and diets of fresh vegetables, fruit and raw meat (i.e., hunter-gatherer lifestyles). Paleolithic ancestors existed in calorie-limited environments and were adapted to minimize energy expenditure wherever possible and maximise consumption of fat, sugar and salt driven by cravings. Paleolithic ancestors existed in outdoor environments with direct exposure to microorganisms and parasites through contact with soil, plants, and animals resulting in a diversity of microorganisms thriving on the skin and within the gastrointestinal tract (i.e., the human microbiota) [119]. The symbiotic relationship between the human microbiota, acquired after birth, and the immune system drives protective responses to pathogens and tolerance to innocuous antigens.

In contrast, modern urban dwellers exist in towns and cities and have lifestyles that are indoors and involve consumption of over-processed foods, and prescription and recreational drugs. Modern urban lifestyles are becoming increasingly sedentary with excessive amounts of time being physically inactive leading to ‘disuse syndromes’ (e.g., ‘walking deficiency syndrome’, ‘hyper-sitting syndrome’) and an increased risk of non-specific chronic musculoskeletal pain [120,121,122]. The shift to urban living is associated with a decline nutritional diversity due to consumption of fewer vegetables, fruits, antioxidants and omega-3 fatty acids and a rise in consumption of high calorie over-processed foods creating diet-induced proinflammatory states [123]. The gastrointestinal microbiota–brain axis of modern humans has been disrupted by the shift to indoor living with environmentally controlled air, sanitation, processed foodstuffs, and the use of pharmaceuticals to eradicate infections and parasites. Reduced exposure to microorganisms in childhood results in immune systems unable to differentiate pathogens that confer benefit or harm and this may contribute to inappropriate immune (allergic) responses to harmless allergens associated with a disrupted microbiota resulting in inflammatory responses and peripheral and central sensitization of the nociceptive system [124]. Evidence suggests that abnormal immune responses may directly influence processes associated with sensitization of the peripheral and central nociceptive system through atypical regulation and output of the hypothalamic–pituitary–adrenal axis including downstream signaling that sensitizes nociceptive afferents [125]. These processes may be contributory factors in visceral pain resulting from inflammatory bowel disease, celiac disease, and metabolic syndrome [119] and idiopathic pain disorders manifesting with persistent central sensitization such as fibromyalgia, chronic pelvic pain, and migraine [125].

Current models of care for chronic pain, especially related to the musculoskeletal system, promote lifestyle adjustment. In essence, they are attempting to rebalance this evolutionary mismatch. Physical activity and diet is at the core of lifestyle adjustment (e.g., for non-specific chronic low back pain [20]). Diet therapies that increase consumption of antioxidants and omega-3 fatty acids, and reduce consumption of exocitoxic substances (e.g., monosodium glutamate, hydrolyzed protein, protein isolates/concentrates, yeast extract, aspartame) and foodstuffs (e.g., bran, nuts, soybean, and aged cheeses) may alleviate chronic pain [126,127,128]. Dietary interventions such as probiotics and prebiotics may prove beneficial to alleviate visceral hypersensitivity associated with a disrupted microbiota [119,129]. However, adherence to long-term lifestyle adjustment is poor and this may be due to the social and environmental conditions in which patients live.

## 9. Upstream for Solutions

Clinical guidelines recommend that practitioners counsel people living with persistent pain to undertake healthy lifestyles. Pain education about the nature of lifestyle adjustments is prominent in pain discourse but incongruous social and environmental conditions may hinder long-term behaviour change. Consideration of upstream solutions may help to reduce chronic pain at both individual and population level.

### 9.1. Social Models of Health Promotion

People with chronic pain prioritize justice-related issues within the context of their personal concerns and needs [130]. Low socioeconomic status, poor working conditions or unemployment, unstable home life, low levels of education, and living in deprived environments are associated with increased pain [108]. People with chronic pain believe that treatment of chronic pain is unfair [131]. Interpretative phenomenological analyses demonstrate that chronic pain patients from lower socioeconomic groups express concerns associated with unfair advantages of others, whereas those from middle socioeconomic groups are concerned with a battle for quality of life, and those from upper socioeconomic groups with the quality of care [130].

Social models of health promotion have been used successfully to tackle social injustices detrimental to health and wellbeing, such as poverty, provision of inexpensive foodstuffs that encourage healthy diets, poor housing and lack of employment [132]. There is however, limited discourse about the use of social models of health promotion within pain literature [133].

### 9.2. Evolutionary-Concordant Environments

There are also constraints of healthy living imposed by the nature of built environments. Often, urban environments are not conducive to undertaking physical activity outdoors due to limited greenspace, fear for personal safety, and air and noise pollution. Indoor activities may introduce additional issues such as transportation to and from a gym, financial expense to use a gym, and chronic overuse injuries such as strains and tendonitis from, for example, jogging on treadmills in restrictive shoes [134]. The Evolutionary Determinants of Health program launched in 2014 provides a framework for debate of evolutionary-concordant healthy cities and social regimes for urban societies, with green, open spaces and clean air to encourage individuals to be physically active [135]. However, there needs to be commitment on the part of policy makers to address social injustices and construct healthy urban environments to enable patients and individuals to live healthy lifestyles.

## 10. Conclusions

Chronic pain remains a major health care problem posing numerous challenges for researchers, practitioners, policy makers and patients alike. In this review, I have used a broad lens to explore a variety of contemporary matters associated with chronic pain. I have argued that pain is perplexing, subjective and amorphous and some patients feel that health care providers do not believe that they have significant pain. I have argued that it is not possible to disprove a person’s pain and that the association between pain and pathology may be tenuous in some circumstances. I described the consequences of long-term prescribing of analgesic medication and offered examples of service delivery that may promote a biopsychosocial model of care at the first point of contact in primary care.

Ultimately, pain is a psychological construct arising from physiological processes occurring in the brain. I have described pain as a top-down perceptual inference that integrates sensory, motor, affective, cognitive, social and environmental contexts to update the final experience of pain. I argue that pain does not faithfully reflect tissue status but serves instead to *infer* bodily threat and drive behaviors to reduce the impact of threat on the integrity of the body. I have provided examples of the malleability of pain perception and offered examples of interventions that manipulate context to alleviate pain. Finally, I have speculated that modern urban environments are painogenic in nature and incompatible with Paleolithic physiology encoded by our genetic heritage. I believe that exploration of pain through the lens of evolutionary mismatch may provide novel insights to why patients have difficulties adhering to healthy lifestyles and provide upstream strategies to reduce the burden of chronic pain on society.

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
