# Peer review of "The Landscape of Chronic Pain: Broader Perspectives"

_medicina, 2019, doi:10.3390/medicina55050182_

Round 1

Reviewer 1 Report

This is a well-written narrative (non-systematic) review focussed on a variety of interesting, contemporary issues related to chronic pain and its management. As noted by the author, the review is ambitious, and the author attempts to cover a lot of territory over relatively few words. A strength of the review is the attention paid to often neglected aspects, including those relating to the multidimensional experience of pain such as multisensory perceptual inferences, and the broader (social and environmental) contexts of chronic pain, particularly the potential role of evolutionary mismatch – these tend to be overlooked in most reviews on chronic pain and are a welcome addition. The review clearly has value and warrants publication, although more about central sensitization, the phenomenon assumed to underlie many centralized pain disorders, and how it relates to the issues discussed, is required, given its importance in this field of study. A few minor revisions would help improve the paper before publication also.

Specific points

Although      the review is not intended to specifically address the neurophysiology of      pain, to the extent that central sensitization is assumed to play a role      in the establishment and maintenance of a number of chronic pain      conditions (Woolf, 2011), it is surprising it receives scant attention in      the present work. At the very least, consideration of the relationship      between central sensitization mechanisms and clearly relevant      concepts/topics addressed, such as the (tenuous) associations between pain      and pathology/injury, the ‘nociplastic’ pain descriptor, and top-down      processing in pain perception (Adams & Turk, 2018), would be helpful      for the reader.

Section      3.1 Disproving pain. The point      summed up in the last sentence of this section, that ‘the subjective nature      of pain makes it is impossible to prove or disprove a person’s pain’ is an      important one, particularly with respect to clinical practice. The point      could be substantiated further with brief reference to recent review work      strongly suggesting that pain malingering cannot be reliably identified by      self-reported questionnaires, physical capacity testing, symptom validity      tests, or clinical examinations (Tuck et al., 2018).

Section 3.2 The      tenuous link between pain and pathology. The line of thought here could be extended further. The assessment of pain      legitimacy is inherently problematic because many of the phenomena      employed in assessment tools (e.g., superficial or nonanatomic tenderness,      effects of distraction) can be explained by advances in pain science (e.g., central sensitization, fear-avoidance behaviour).      Some reference to recent proposals to employ multimodal assessment models      of pain (intended to address the potential alignment between the relative      scopes of pain measures versus pain expression by integrating elements of      patients’ subjective pain experience; Wideman et al., 2018) may be helpful      here.

Section      3.2 The tenuous link between pain      and pathology. Line 120. The statement ‘Pain may      occur is the presence of minor injuries’ should presumably be ‘Pain      may occur in the presence of minor injuries’.

Section 4.4 International Classification of      Diseases: Chronic Primary Pain While      I agree with the author that the introduction of chronic primary pain as a      new diagnostic entity recognizes conditions not satisfactorily accounted for      in categories defined by somatic or psychological etiology, perhaps a note      here could be made that this development in part reflects arguments that pain      is a pathologic entity in its own right (Raffaeli et al., 2017).

Section      5.1 – 5.3. The review briefly addressed the use of analgesic and adjuvant      drugs that modulate nociceptive system processing as a first line      treatment for pain. Given the multidimensional approach to (chronic) pain      adopted throughout, I wonder if (brief) consideration should also be given      to the role of placebo responses in treatment for pain conditions. For      example, open-hidden paradigms have provided experimental evidence showing      that hidden administration of an analgesic treatment significantly reduce      the efficacy of the treatment (Benedetti et al., 2011), suggesting the action of different analgesic      agents can be modulated by cognitive and affective factors. Conversely,      increasing evidence suggests that open label placebo can be a safe and      effective adjunct to treatment for chronic lower back pain (Carvalho et      al., 2016).

Section      7.4 Multisensory perceptual inference to alleviate pain Line 365. The      part of the statement ‘...reducing threat associated moving painful body      parts...’ should presumably be ‘...reducing threat associated with moving      painful body parts...’.

Section      8.3. Chronic pain and      evolutionary mismatch. The author links modern lifestyles with      inappropriate immune (allergic) responses to innocuous antigens and risk      of chronic pain conditions. But accumulating evidence suggests that      abnormal immune responses may directly impact on pain sensitization      processes, which should be addressed here too. For example, abnormal regulation and      output of the hypothalamic-pituitary-adrenal (HPA) axis is commonly      associated with centralized pain disorders such as fibromyalgia, chronic pelvic pain, and migraine (altered      downstream signalling from the HPA axis can lead to sensitization of      nearby nociceptive afferents; see Eller-Smith et al., 2018).

References

Adams LM, Turk DC. Central sensitization and the biopsychosocial approach to understanding pain. Journal of Applied Biobehavioral Research. 2018 Jun;23(2):e12125.

Benedetti F, Carlino E, Pollo A. Hidden administration of drugs. Clinical Pharmacology & Therapeutics. 2011 Nov;90(5):651-61.

Carvalho C, Caetano JM, Cunha L, Rebouta P, Kaptchuk TJ, Kirsch I. Open-label placebo treatment in chronic low back pain: a randomized controlled trial. Pain. 2016 Dec;157(12):2766.

Eller-Smith OC, Nicol AL, Christianson JA. Potential mechanisms underlying centralized pain and emerging therapeutic interventions. Frontiers in Cellular Neuroscience. 2018 Feb 13;12:35.

Raffaeli W, Arnaudo E. Pain as a disease: an overview. Journal of Pain Research 2017;

10:2003–8.

Tuck NL, Johnson MH, Bean DJ. You'd Better Believe It: The conceptual and practical challenges of assessing malingering in patients with chronic pain. The Journal of Pain. 2019 Feb 1;20(2):133-45.

Wideman TH, Edwards RR, Walton DM, Martel MO, Hudon A, Seminowicz DA. The Multimodal Assessment Model of Pain: a novel framework for further integrating the subjective pain experience within research and practice. The Clinical Journal of Pain. 2019 Mar 1;35(3):212-21.

Woolf, C. J. (2011). Central sensitization: Implications for the diagnosis and treatment of pain. Pain, 152(3 Suppl), S2–S15.

Author Response

Reviewer 1

This is a well-written narrative (non-systematic) review focussed on a variety of interesting, contemporary issues related to chronic pain and its management. As noted by the author, the review is ambitious, and the author attempts to cover a lot of territory over relatively few words. A strength of the review is the attention paid to often neglected aspects, including those relating to the multidimensional experience of pain such as multisensory perceptual inferences, and the broader (social and environmental) contexts of chronic pain, particularly the potential role of evolutionary mismatch – these tend to be overlooked in most reviews on chronic pain and are a welcome addition. The review clearly has value and warrants publication, although more about central sensitization, the phenomenon assumed to underlie many centralized pain disorders, and how it relates to the issues discussed, is required, given its importance in this field of study. A few minor revisions would help improve the paper before publication also.

Author’s Response: Thank you for taking the time to critique my ambitious and eclectic the review. I agree with all of your comments and have attempted to resolve each of them in turn. I have attempted to contain the length of the revisions and hope that this meets with your approval.

Specific points

Although the review is not intended to specifically address the neurophysiology of pain, to the extent that central sensitization is assumed to play a role in the establishment and maintenance of a number of chronic pain  conditions (Woolf, 2011), it is surprising it receives scant attention in  the present work. At the very least, consideration of the relationship between central sensitization mechanisms and clearly relevant  concepts/topics addressed, such as the (tenuous) associations between pain  and pathology/injury, the ‘nociplastic’ pain descriptor, and top-down  processing in pain perception (Adams & Turk, 2018), would be helpful  for the reader.

Author’s Response: I agree and thank you for alerting me to the Journal of Applied Biobehavioral Research’s Special Issue on Central Sensitization – an excellent resource!  I have added a section on the basic principles and mechanisms of sensitization to section 4.1. I have also added statements throughout the article to draw on aspects central sensitisation relevant to the discussions.

Section 3.1 Disproving pain. The point summed up in the last sentence of this section, that ‘the subjective nature of pain makes it is impossible to prove or disprove a person’s pain’ is an  important one, particularly with respect to clinical practice. The point could be substantiated further with brief reference to recent review work strongly suggesting that pain malingering cannot be reliably identified by self-reported questionnaires, physical capacity testing, symptom validity tests, or clinical examinations (Tuck et al., 2018).

Author’s Response: Thank you for alerting me to this recent article. I have included a sentence about it to this section.

Section 3.2 The tenuous link between pain and pathology. The line of thought here could be extended further. The assessment of pain legitimacy is inherently problematic because many of the phenomena employed in assessment tools (e.g., superficial or nonanatomic tenderness, effects of distraction) can be explained by advances in pain science (e.g., central sensitization, fear-avoidance behaviour).  Some reference to recent proposals to employ multimodal assessment models of pain (intended to address the potential alignment between the relative  scopes of pain measures versus pain expression by integrating elements of  patients’ subjective pain experience; Wideman et al., 2018) may be helpful here.

Author’s Response: I have added a few sentences to enrich the discussion of these points this section 3.2. Once again, thank you for alerting me to the article by Wideman and I have included a discussion of it in section 3.3.

Section  3.2 The tenuous link between pain and pathology. Line 120. The statement ‘Pain may      occur is the presence of minor injuries’ should presumably be ‘Pain may occur in the presence of minor injuries’.

Author’s Response: Amended

Section 4.4 International Classification of Diseases: Chronic Primary Pain While      I agree with the author that the introduction of chronic primary pain as a new diagnostic entity recognizes conditions not satisfactorily accounted for in categories defined by somatic or psychological etiology, perhaps a note here could be made that this development in part reflects arguments that pain is a pathologic entity in its own right (Raffaeli et al., 2017).

Author’s Response: Of course. I have made reference to Raffaeli’s review in section 4.4.

Section 5.1 – 5.3. The review briefly addressed the use of analgesic and adjuvant      drugs that modulate nociceptive system processing as a first line treatment for pain. Given the multidimensional approach to (chronic) pain adopted throughout, I wonder if (brief) consideration should also be given      to the role of placebo responses in treatment for pain conditions. For example, open-hidden paradigms have provided experimental evidence showing  that hidden administration of an analgesic treatment significantly reduce the efficacy of the treatment (Benedetti et al., 2011), suggesting the action of different analgesic agents can be modulated by cognitive and affective factors. Conversely, increasing evidence suggests that open label placebo can be a safe and effective adjunct to treatment for chronic lower back pain (Carvalho et al., 2016).

Author’s Response: I agree the work of Benedetti and others in this field is fascinating and I have added a paragraph to section 5.1 to reflect the issue

Section 7.4 Multisensory perceptual inference to alleviate pain Line 365. The part of the statement ‘...reducing threat associated moving painful body  parts...’ should presumably be ‘...reducing threat associated with moving painful body parts...’.

Author’s Response: Amended

Section 8.3. Chronic pain and      evolutionary mismatch. The author links modern lifestyles with inappropriate immune (allergic) responses to innocuous antigens and risk      of chronic pain conditions. But accumulating evidence suggests that      abnormal immune responses may directly impact on pain sensitization  processes, which should be addressed here too. For example, abnormal regulation and      output of the hypothalamic-pituitary-adrenal (HPA) axis is commonly      associated with centralized pain disorders such as fibromyalgia, chronic pelvic pain, and migraine (altered      downstream signalling from the HPA axis can lead to sensitization of      nearby nociceptive afferents; see Eller-Smith et al., 2018).

Author’s Response: Again, thank you for this suggestion and I have included further clarity with reference to the excellent review by Eller-Smith

References

Adams LM, Turk DC. Central sensitization and the biopsychosocial approach to understanding pain. Journal of Applied Biobehavioral Research. 2018 Jun;23(2):e12125.

Benedetti F, Carlino E, Pollo A. Hidden administration of drugs. Clinical Pharmacology & Therapeutics. 2011 Nov;90(5):651-61.

Carvalho C, Caetano JM, Cunha L, Rebouta P, Kaptchuk TJ, Kirsch I. Open-label placebo treatment in chronic low back pain: a randomized controlled trial. Pain. 2016 Dec;157(12):2766.

Eller-Smith OC, Nicol AL, Christianson JA. Potential mechanisms underlying centralized pain and emerging therapeutic interventions. Frontiers in Cellular Neuroscience. 2018 Feb 13;12:35.

Raffaeli W, Arnaudo E. Pain as a disease: an overview. Journal of Pain Research 2017;

10:2003–8.

Tuck NL, Johnson MH, Bean DJ. You'd Better Believe It: The conceptual and practical challenges of assessing malingering in patients with chronic pain. The Journal of Pain. 2019 Feb 1;20(2):133-45.

Wideman TH, Edwards RR, Walton DM, Martel MO, Hudon A, Seminowicz DA. The Multimodal Assessment Model of Pain: a novel framework for further integrating the subjective pain experience within research and practice. The Clinical Journal of Pain. 2019 Mar 1;35(3):212-21.

Woolf, C. J. (2011). Central sensitization: Implications for the diagnosis and treatment of pain. Pain, 152(3 Suppl), S2–S15.

Reviewer 2 Report

In his paper entitled “The landscape of Chronic Pain: Broader Perspectives”, the author M. Jonhson refreshed and dusted many notions and beliefs about chronic pain experience in humans. This essay is well written, pleasant to read, and opens interesting philosophical perspectives for the reader. However, the reviewer advice to revise / moderate some generalization that are sometimes too anthropomorphic in the philosophical approach.

In fact, many vertebrate species can experience chronic pain. These animals will also express their chronic discomfort, in a behaviorally relevant language to their own species filtered by their own evolutive or environmental motivations.

It is therefore of importance to clearly state that the author's personal views and opinions reported here cannot be generalized. Thus, in rodents (i.e. mouse, rat, which is a social animal family particularly well characterized for physiology, behavioral expression and impact on the quality of life of chronic pain), most of the arguments presented by the author cannot be applied. Each theory deserves an anti-thesis and a philosophical deepening could bring the reader in deeper thinking. Thus, the philosophical part could be a little more controversial.

In part 2.1., the author states: “This is because pain is a psychological state and not activity in nociceptive (noxious detecting) pathways”

I think this is an un-appropriate statement that neglect the real physiological support of most (all) of chronic pain states. I truly believe that because we cannot yet identify the physiological source of a chronic pain state (because of technical difficulties that we have not yet challenge), we cannot embed under the psychological umbrella these discomforts. IBS and fibromyalgia are striking examples. Patients affected by those conditions are awaiting delayed medical responses and discredit the possible physiological causes of their affliction seem a little disrespectful, and scientifically nearly medieval.

Modern medicine is still a young discipline, and many discoveries are to come. I think that our interpretation of chronic pain with the stammering of answers that the science has only given us at the moment must be done more humbly, surrounded by more precaution.

Part 2.3. Why do we communicate pain?

You elaborate how expression of pain may be justify by generation of empathy, but this is only relevant to our modern society. Empathy haven’t been an evolutionary motor. Only decades (century) ago, pain expression was consider as a expression of weakness. I cannot even imagine how our very close ancestor such as cro-magnon may have treat member of the clan un-capable to contribute to the survival of the group. Survival always have surpassed (overcome) empathy. Even nowadays, empathy is only expressed when no life-threatening condition are extinguishing it. Our society has totally abolished our survival needs and it is only very recently that empathy has start to be expressed. While the point of view of the author are currently pertinent, they are only in regard of our current, western society. There is not back pain epidemic in country with alimentary pressure. Not that people do not suffer, but there is simply no point to over-express it.

I doubt our paleolithic ancestors may a have sick for empathy from the rest of the group. Showing weakness would have more likely turn short and results in abrupt, final and definitive pain relieve.

Whether my point is more or less true that the point of view developed by the author, it is as valid. I would not be surprise that a clan member with chronic pain would have simply isolate himself in an altruist fashion to not weaken the overall group. Altruism is a behavioral trait that is likely evolutionary more ancient than ampathy (and likely share some phylogeny with empathy) and is shared cross various species including non-vertebrae and non-mammalian social species such as ants or some birds.

I do not see how Evolution could have possibly hardwire expression of chronic pain. Evolution is not only the 100 last years. Evolution take place over hundreds of generations, and humans were likely much less heart soft centuries ago. They had survival pressure only few centuries ago.

Finally, to come back to animal species that express pain, even non-social animal express behavioral sign of pain. While chronic pain is more challenging to observe in most animals, acute and sub-acute pain is unanimously express across many specie with shared behavioral trait such as facial expression and specific body position. The many grimace scales recently develop are largely taking advantage of it.

As a complementary thought, in our modern society, pain management rely almost exclusively upon pain expression capacity. I think his notion is actually missing in the present assay and represent another important aspect of the problematic developed here.

Indeed, people with altered cognitive capacity receive very poor analgesic support.

A striking example are patients with Alzheimer's disease. These patient, who are afflicted with as many chronic painful conditions than age-match people receive however significantly less analgesic prescription (and almost exclusively paracetamol). However, this is not because a patient does not express any discomfort that we can be sure that no discomfort is experienced.

Part 4.2. Nociplastic pain:

I think this part should be more moderate and balanced. By over-using nociplastic pain theory, we might ultimately under-investigate for somatic origin of nociceptive signal. While in some exceptional case it is out of doubt that pain is centrally driven, in vast majority of chronic pain conditions, we should not stop investigating for somatic disorder, keeping in mind that the primary role of pain is a warming signal, a symptom associated with abnormal physiological condition.

5.3. novel centrally acting drugs

Given the limited record and published data in the field, it is difficult to decide positively on the usefulness of the use of cannabis derivatives. FYI, a review has just been published on the subject suggesting a benefit in the management of chronic pain. Urits, Borchart, Hasegawa, Kochanski, Orhurhu, Viswanath. An Update of Current Cannabis-Based Pharmaceuticals in Pain Medicine. Pain Ther. 2019

Major positive points:

I particularly appreciate the parts 2.2. and 8. In part 2.2. I found inspiring to include capacity to perceive and envision past, present and future that could be further process to explain/interpret specie-specific pain expression and behavioral adaptation. The discussion on Social and Environmental contexts (part 8), although flying over the subject, opens many interesting avenues of reflection.

Author Response

Reviewer 2

In his paper entitled “The landscape of Chronic Pain: Broader Perspectives”, the author M. Jonhson refreshed and dusted many notions and beliefs about chronic pain experience in humans. This essay is well written, pleasant to read, and opens interesting philosophical perspectives for the reader. However, the reviewer advice to revise / moderate some generalization that are sometimes too anthropomorphic in the philosophical approach.

In fact, many vertebrate species can experience chronic pain. These animals will also express their chronic discomfort, in a behaviorally relevant language to their own species filtered by their own evolutive or environmental motivations.

Author’s Response: Thank you for taking the time to provide thought provoking comments, especially on matters related to non-human expression of what we believe to be pain. I hope that I have interpreted your comments correctly. I have attempted to address all of your concerns. I have resisted the urge to expand the length of the article too much and hope that this meets with your approval.

It is therefore of importance to clearly state that the author's personal views and opinions reported here cannot be generalized. Thus, in rodents (i.e. mouse, rat, which is a social animal family particularly well characterized for physiology, behavioral expression and impact on the quality of life of chronic pain), most of the arguments presented by the author cannot be applied. Each theory deserves an anti-thesis and a philosophical deepening could bring the reader in deeper thinking. Thus, the philosophical part could be a little more controversial.

Author’s Response: I appreciate this point and have included a statement in the statement of the intention of the review (Introduction) to reflect that some arguments are anthropomorphic in nature and cannot be generalized to non-human species.

 In part 2.1., the author states: “This is because pain is a psychological state and not activity in nociceptive (noxious detecting) pathways” I think this is an un-appropriate statement that neglect the real physiological support of most (all) of chronic pain states. I truly believe that because we cannot yet identify the physiological source of a chronic pain state (because of technical difficulties that we have not yet challenge), we cannot embed under the psychological umbrella these discomforts. IBS and fibromyalgia are striking examples. Patients affected by those conditions are awaiting delayed medical responses and discredit the possible physiological causes of their affliction seem a little disrespectful, and scientifically nearly medieval. Modern medicine is still a young discipline, and many discoveries are to come. I think that our interpretation of chronic pain with the stammering of answers that the science has only given us at the moment must be done more humbly, surrounded by more precaution.

Author’s Response: Thank you for identifying this – on re-reading I see that my original statement was far more dogmatic than I had intended. I have revised the sentence and added some further clarity.

Part 2.3. Why do we communicate pain?

You elaborate how expression of pain may be justify by generation of empathy, but this is only relevant to our modern society. Empathy haven’t been an evolutionary motor. Only decades (century) ago, pain expression was consider as a expression of weakness. I cannot even imagine how our very close ancestor such as cro-magnon may have treat member of the clan un-capable to contribute to the survival of the group. Survival always have surpassed (overcome) empathy. Even nowadays, empathy is only expressed when no life-threatening condition are extinguishing it. Our society has totally abolished our survival needs and it is only very recently that empathy has start to be expressed. While the point of view of the author are currently pertinent, they are only in regard of our current, western society. There is not back pain epidemic in country with alimentary pressure. Not that people do not suffer, but there is simply no point to over-express it.

I doubt our paleolithic ancestors may a have sick for empathy from the rest of the group. Showing weakness would have more likely turn short and results in abrupt, final and definitive pain relieve. Whether my point is more or less true that the point of view developed by the author, it is as valid. I would not be surprise that a clan member with chronic pain would have simply isolate himself in an altruist fashion to not weaken the overall group. Altruism is a behavioral trait that is likely evolutionary more ancient than ampathy (and likely share some phylogeny with empathy) and is shared cross various species including non-vertebrae and non-mammalian social species such as ants or some birds.

Author’s Response: I appreciate the excellent points that you have raised here, although I do not entirely agree with the premise that empathy is only relevant to modern society (apologies if I have misunderstood your point). Non-human species express behavior’s that may be proxy empathy suggesting that it may have evolutionary roots (research by Jeffery Mogil is of relevance). There is certainly a debate to be had about the role of empathy in human and non-animals and how this interfaces with chronic pain. I agree that I needed to be less dogmatic in my original statements and have followed your advice by revising the content and tone of section 1.3 to highlight the speculative nature of my arguments

I do not see how Evolution could have possibly hardwire expression of chronic pain. Evolution is not only the 100 last years. Evolution take place over hundreds of generations, and humans were likely much less heart soft centuries ago. They had survival pressure only few centuries ago.

Finally, to come back to animal species that express pain, even non-social animal express behavioral sign of pain. While chronic pain is more challenging to observe in most animals, acute and sub-acute pain is unanimously express across many specie with shared behavioral trait such as facial expression and specific body position. The many grimace scales recently develop are largely taking advantage of it.

Author’s Response: Again, thank for some excellent insights. I agree that ‘…. humans were likely much less heart soft centuries ago …’ but I would not go so far as to say that there were no selective pressures to express pain in the presence of persistent sensitisation and hyperalgesia. There are excellent articles on the potential evolutionary role of sensitization and hyperalgesia in chronic pain {Walters, 1994 #55915;Price, 2014 #55914;Williams, 2016 #51887;Mogil, 2015 #55916}. I have revised section 8.2 to reflect your point and have included the references listed above.

As a complementary thought, in our modern society, pain management rely almost exclusively upon pain expression capacity. I think his notion is actually missing in the present assay and represent another important aspect of the problematic developed here. Indeed, people with altered cognitive capacity receive very poor analgesic support.  A striking example are patients with Alzheimer's disease. These patient, who are afflicted with as many chronic painful conditions than age-match people receive however significantly less analgesic prescription (and almost exclusively paracetamol). However, this is not because a patient does not express any discomfort that we can be sure that no discomfort is experienced.

Author’s Response: Thank you again for another point well made. I have amended section 3.1 and included a brief discussion on challenges facing those that are unable to offer verbal expression of their internal states and the importance of multimodal assessment including behavioral observation

Part 4.2. Nociplastic pain:

I think this part should be more moderate and balanced. By over-using nociplastic pain theory, we might ultimately under-investigate for somatic origin of nociceptive signal. While in some exceptional case it is out of doubt that pain is centrally driven, in vast majority of chronic pain conditions, we should not stop investigating for somatic disorder, keeping in mind that the primary role of pain is a warming signal, a symptom associated with abnormal physiological condition.

Author’s Response: I have emphasized this point by adding text in section 4.2

5.3. novel centrally acting drugs

Given the limited record and published data in the field, it is difficult to decide positively on the usefulness of the use of cannabis derivatives. FYI, a review has just been published on the subject suggesting a benefit in the management of chronic pain. Urits, Borchart, Hasegawa, Kochanski, Orhurhu, Viswanath. An Update of Current Cannabis-Based Pharmaceuticals in Pain Medicine. Pain Ther. 2019

Author’s Response: I agree. The review that I cited to support this sentence provides evidence for this. Thank you for the additional reference which I found most interesting and I have added it to the section.  

Major positive points:

I particularly appreciate the parts 2.2. and 8. In part 2.2. I found inspiring to include capacity to perceive and envision past, present and future that could be further process to explain/interpret specie-specific pain expression and behavioral adaptation. The discussion on Social and Environmental contexts (part 8), although flying over the subject, opens many interesting avenues of reflection.

Author’s Response: Thank you for your careful consideration and excellent critique of my review

Round 2

Reviewer 2 Report

very nice update of your essay. thanks for integrating so well most of my comments. Brave work!